



# Speciated atmospheric mercury at Waliguan Global Atmospheric Watch station in the northeastern Tibetan Plateau: implication of dust related sources for particulate bound mercury

Hui Zhang[1], Xuewu Fu[1,2*], Ben Yu[3], Baoxin Li[4], Peng Liu[4], Guoqing Zhang[4], Leiming Zhang[5], Xinbin Feng[1,2,6*]

[1] State Key Laboratory of Environmental Geochemistry, Institute of Geochemistry, Chinese Academy of Sciences, 99 Lincheng West Road, Guiyang, 550081, China.

[2] CAS Center for Excellence in Quaternary Science and Global Change, Xi'an, 710061, China.

[3] State Key Laboratory of Environmental Chemistry and Ecotoxicology, Research Center for Eco-Environmental Sciences, Chinese Academy of Sciences, Beijing, 100085, China.

[4] China Global Atmosphere Watch Baseline Observatory, Qinghai Meteorological Bureau, Xining, 810001, China.

[5] Air Quality Research Division, Science and Technology Branch, Environment and Climate Change Canada, Toronto, M3H5T4, Canada

[6] University of Chinese Academy of Sciences, Beijing, 100049, China.

Corresponding authors:

Xuewu Fu (fuxuewu@mail.gyig.ac.cn), Xinbin Feng (fengxinbin@vip.skleg.cn)



**Abstract**
To understand the ambient levels and sources of atmospheric mercury (Hg) in the Tibetan
Plateau, a full-year continuous measurement of speciated atmospheric mercury was conducted at
Waliguan (WLG) Baseline Observatory (3816 m a.s.l.) from May 2012 to April 2013. Mean
concentrations (±1SD) of gaseous elemental mercury (GEM), gaseous oxidized mercury (GOM)
and particulate bound mercury (PBM) during the whole study period were $1.90 \pm 0.80$ ng m$^{-3}$, 12.0
$\pm$ 10.6 pg m$^{-3}$ and $65.4 \pm 63.2$ pg m$^{-3}$, respectively. Seasonal variations of GEM were very small,
while those of PBM were quite large with mean values being four times higher in cold ($102.3 \pm 66.7$
pg m$^{-3}$) than warm ($22.8 \pm 14.6$ pg m$^{-3}$) season. Anthropogenic emissions to the east of Tibetan
Plateau contributed significantly to GEM pollution at WLG, while dust particles originated from
desert and Gobi regions in Xinjiang province and Tibetan Plateau to the west of WLG were
responsible to PBM pollution at WLG. This finding is also supported by the significant positive
correlation between daily PBM concentration and daily cumulative absorbing aerosol index (AAI)
encountered by air masses transported during the preceding two days.
**Keywords:** Speciated atmospheric mercury, Particulate bound mercury, Anthropogenic mercury
emissions, Dust related sources
**Introduction**
Mercury (Hg) is a toxic pollutant of global concern due to its long lifetime in air,
bioaccumulation in aquatic system, and detrimental impacts on human and animal health.
Atmospheric Hg is operationally defined in three forms, i.e., gaseous elemental mercury (GEM),
gaseous oxidized mercury (GOM), and particulate bound mercury (PBM). These Hg species can be
transformed among each other through complex physical and chemical processes (Lyman et al.,
2020; Selin, 2009). For example, GEM can be oxidized to form GOM, and GEM and/or GOM can
adsorb on atmospheric aerosols to form PBM. GEM has a lifetime in air of 0.5-2 years while GOM
and PBM only have a lifetime of hours to weeks (Ariya et al., 2015; Murphy et al., 2006). Because
of their different lifetimes, GEM can be transported globally via atmospheric circulation whereas
PBM is limited to regional transport (Pirrone et al., 2010). These Hg species can be removed from
the atmospheric through dry and wet deposition processes. Once deposited to earth's surface, Hg
can be converted to methylmercury by biological processes, which can cause potential risks to
ecological and human health (Jonsson et al., 2014; Wright et al., 2018). On the other hand, Hg
accumulated in soil and water bodies can be emitted into the atmosphere, which plays an important
role in the global atmospheric Hg cycle (Obrist et al., 2018; Wang et al., 2016).



Spatiotemporal variations of atmospheric Hg and Hg speciation fractions within the total Hg
are controlled by many factors, among which anthropogenic emissions are important ones (Driscoll
et al., 2013; Fu et al., 2012b). Global anthropogenic Hg emissions to the atmosphere were estimated
to be 2224 Mg yr$^{-1}$, of which 82-87%, 10-18% and 3-4% were in the form of GEM, GOM and PBM,
respectively (AMAP/UNEP 2018). In China, concentrations of GEM and PBM were generally
elevated compare to the observations in Europe and America. The fractions of PBM in total
atmospheric Hg in urban areas of China were in the range of 5.2–17.2%, higher than those from
total anthropogenic Hg emissions (Fu et al., 2015; Zhang et al., 2015b). In addition, observations of
speciated atmospheric Hg in China in both urban and rural areas of China showed generally higher
PBM than GOM levels (Fu et al., 2015b), which is in contrast with the higher GOM than PBM
fractions in the total anthropogenic Hg emissions in China (Zhang et al., 2015b). These findings
indicate that additional emission sources and other physical and chemical processes contributed to
the elevated PBM concentrations in China. For example, natural sources, such as biomass burning
and dust related sources, and gas-particle partitioning can also produce PBM (Amos et al., 2012;
Obrist et al., 2008). The impact of these sources and processes on atmospheric PBM, although is
potentially important, has not been well investigated by previous studies.

Speciated atmospheric Hg has been monitored in China in recent years, but observations in the
Tibetan Plateau region are very limited. The Tibetan Plateau, also known as the third pole of the
world with an average altitude of over 4000 m a.s.l., is an ideal place for assessing transport and
transformation of atmospheric pollutants in China and other Asian regions (Chen et al., 2019;
Loewen et al., 2007). The Tibetan Plateau is surrounded by East Asia and South Asia, which are the
two most important source regions of atmospheric Hg in the world (Zhao et al., 2013). The
Taklamakan and Gobi deserts are located to the west and north, respectively, of the Tibetan Plateau.
Since the Tibetan Plateau region is strongly impacted by westerlies, Indian Summer Monsoon and
East Asia Summer Monsoon (Fig.1), it is affected by air pollutants from anthropogenic and natural
source regions adjacent to the Tibetan Plateau (Che et al., 2011). Previous studies postulated that air
masses passing over the urban and industrial areas in western China and South Asia were important
sources of atmospheric GEM at Waliguan and Nam Co (4730 m a.s.l.) on the northeastern edge and
Midlands of the Tibetan Plateau, respectively (Fu et al., 2012a; Yin et al., 2018). At Shangri-La,
located on the southeastern edge of the Tibetan Plateau, the identified atmospheric GEM source
regions were also located in Southeast Asia and mainland China (Zhang et al., 2015a). Speciated
atmospheric Hg measurement was only carried out during a warm season in Qomolangma Natural
Nature Preserve (4276 m a.s.l.) of Tibetan Plateau (Lin et al., 2019). Low GEM concentrations
(means: 1.33 to 1.42 ng m$^{-3}$), but elevated PBM concentrations (means: 25.6 to 49.0 pg m$^{-3}$), were



observed in middle and southern Tibetan Plateau as compared to those in rural central and eastern
China (Yin et al., 2018; Lin et al., 2019). To date, long-term observations of speciated atmospheric
Hg in the Tibetan Plateau region are still lacking, limiting our capacity to fully understand the
spatiotemporal patterns of Hg in this region and associated source regions and controlling factors.

In this study, one-year continuous monitoring of speciated atmospheric Hg was carried out at
WLG in Tibetan Plateau region. Data were analyzed carefully for exploring the ambient levels,
seasonal and diurnal patterns, and source regions of speciated atmospheric Hg in this region.
Knowledge generated from this study is needed for establishing future emission control policies in
order to preserve many sensitive ecosystems in this region.

**2 Materials and methods**
**2.1 Measurement site**
The measurement site is situated at the summit of Mt. Waliguan located at northeastern edge
of the Tibetan Plateau in northwest China. It is a station known as Waliguan (WLG) Baseline
Observatory (100°54' E, 36°17' N, 3816 m a.s.l.) (Fig. 1), which is the only station in inner Asia in
the Global Atmospheric Watch (GAW) program of World Meteorological Organization (WMO).
This area has a typical high plateau continental climate, and western winds dominant at the site (Fu
et al., 2012a; Okamoto and Tanimoto, 2016; Xu et al., 2018). WLG is mainly surrounded by the arid
and semi-arid grassland and desert lands. The population density is very low and industrial activities
sparsely distributed within 80 km around WLG. Anthropogenic Hg emissions in Qinghai province
is relatively low, and are mostly located to the east of WLG (Fu et al., 2015; Sun et al., 2020; Wu
et al., 2006). The Taklimakan Desert and Gobi Desert of Xinjiang province are located to the west
of WLG, and the Gobi Desert of Hexi Corridor and southern Inner Mongolia are located to the north
of WLG (Fig.1).

**2.2 Sampling method**
**2.2.1 Measurements of speciated atmospheric mercury**
High-temporal resolution measurements of GEM, GOM and PBM were carried out using the
2537B-1130-1135 Atmosphere Speciation Mercury Analysis System (Fig. 1, Tekran Inc., Toronto,
Canada) from May 2012 to April 2013. The Tekran Model 2537B Mercury Vapour Analyzer
provides continuous analysis of GEM in air at 0.1 ng m$^{-3}$ detection limit. The instrument samples
air and captures vapour phase Hg on the cartridges containing ultra-pure gold adsorbent media. The
amalgamated Hg is thermally desorbed and detected using Cold Vapour Atomic Fluorescence
Spectrometry (CVAFS). The Model 1135 Particulate Mercury Unit, together with the Model 1130





Mercury Speciation Unit, allows the Model 2537B Mercury Vapor Analyzer to simultaneously
monitor and differentiate between GEM, GOM and PBM (fine fraction, < 2.5 um) in ambient air.
KCl-coated annular denuders was installed in the specially designated location of Model 1130
Mercury Speciation Unit before the instrument starts running. The instrument's workflow is
controlled by the controller which is capable of executing an automatic sampling and analysis
program (Feng et al., 2000; Fu et al., 2016; Lindberg et al., 2002). This system has been used to
monitor atmospheric Hg species worldwide, including the North America Atmospheric Mercury
Network (AMNet) and the Global Mercury Observation System (GMOS) (Lan et al., 2012;
Sprovieri et al., 2016). In this study, data QA/QC procedure followed the GMOS Standard Operation
Procedure and Data Quality Management (D'Amore et al., 2015). Although KCl-coated annular
denuders have been the most popular and widely applied method for measuring ambient GOM,
large analytical uncertainties in GOM may exist due to the trace level and complicated chemical
compounds of GOM which may not be fully collected by denuders (Ariya et al., 2015; Cheng and
Zhang, 2017; Gustin et al., 2015; Gustin et al., 2019). Analysis and discussions presented in this
study are mostly focused on GEM and PBM, considering the larger uncertainties in GOM than GEM
and PBM.

**2.2 Meteorological data and backward trajectory calculation**
Meteorological parameters, including air temperature (AT), relative humidity (RH), rainfall
(RF), wind direction (WD) and wind speed (WS) were obtained from the local weather station at
WLG. In order to identify the effect of long-range transport of Hg emissions on the distributions of
atmospheric Hg at WLG, backward trajectories arriving the site at 100 m above the ground were
calculated every 4 hours using the TrajStat software and gridded meteorological data from the Air
Resource Laboratory, National Oceanic and Atmospheric Administration (NOAA) (Wang et al.,
2009). To investigate the source regions potentially influencing GEM and PBM concentrations at
WLG, a weighing algorithm based on measured concentrations, known as the concentration
weighted trajectory (CWT) approach, was applied in this study. In this procedure, the CWT value
indicates the source strength of a $0.5° × 0.5°$ grid cell ($CWT_{ij}$) to the WLG and is defined as:
$$C_{ij} = \frac{1}{\sum_{l=1}^{M} \tau_{ijl}} \sum_{l=1}^{M} C_l \tau_{ijl}$$

where $C_{ij}$ is the average CWT value of speciated atmospheric Hg in the grid cell ($i,j$), $C_l$ is the
measured Hg concentration at WLG, $\tau_{ijl}$ is the number of trajectory endpoints in the grid cell ($i,j$)
associated with the $C_l$ sample, and $M$ is the number of samples that have trajectory endpoints in grid
cell ($i,j$). A point filter is applied as the final step of CWT to eliminate grid cells with few endpoints.
Weighted concentration fields show concentration gradients across potential sources. This method




helps determine the relative significance of potential source regions (Cheng et al., 2013; Zhang et
al., 2016).

**2.3 Ancillary parameters and analysis**

Anthropogenic emissions of GEM and PBM in 0.5° × 0.5° grid cells in the studied domain

were obtained from the 2010 global emission dataset developed by the Arctic Monitoring and
Assessment program (AMAP) (AMAP/UNEP, 2013). Gridded monthly biomass burned areas at
0.25° spatial resolution were obtained from the fourth version of the Global Fire Emission Database
(GFED4) (Giglio et al., 2013). Absorbing Aerosol Index (AAI) constitutes one of the most useful
space-borne data sets, offering temporal and spatial information on UV absorbing aerosols (black
carbon, desert dust) distributions. Desert dust and biomass burning related aerosols are the dominant
aerosol types detected by the AAI, and AAI is therefore a useful parameter for qualitatively
identifying the dust and biomass burning related sources. The AAI data are available on daily and
monthly basis at a spatial resolution of 1×1 degree. Generally, non-absorbing aerosols (e.g., sulfate
and sea-salt) yield negative AAI values, UV-absorbing aerosols (e.g., dust and smoke) yield positive
AAI values, and clouds yield near-zero values (Prospero et al., 2002). Such information can be used
for identifying distinct desert dust aerosol sources and analyzing dust and smoke transport patterns
(Chiapello et al., 1999; Kubilay et al., 2005; Moulin and Chiapello, 2004). A detailed description of
the AAI product is given in (Herman et al., 1997; Torres et al., 1998). In this study, Global monthly
gridded (1×1 degrees) AAI products during our study period were obtained from the Tropospheric
Emission Monitoring Internet Service (TEMIS) (http://www.temis.nl/airpollution/absaai/).

To study the effect of dust related sources on the variations in PBM concentration at WLG, we

calculated the daily cumulative AAI (ΣAAI) based on the 2-day backward trajectory and gridded
AAI data. Further analysis between the daily ΣAAI and mean PBM concentration were conducted
to assess the effect of dust related sources on the variations of PBM at WLG.

**3 Results and discussion**
**3.1 Concentrations of GEM, GOM and PBM**

Time series of hourly speciated atmospheric Hg concentrations is shown in Fig.2. Mean (±1sd)

concentrations of GEM, GOM and PBM at WLG during the whole sampling campaign were 1.85 ±
0.96 ng m$^{-3}$, 14.0 ± 13.2 pg m$^{-3}$ and 68.1 ± 70.3 pg m$^{-3}$, respectively. Mean GEM level at WLG was
relatively higher than the background levels in the Northern Hemisphere (1.5-1.7 ng m$^{-3}$) (Fu et al.,
2015; Sprovieri et al., 2016). Mean GEM concentration at WLG was relatively lower than that
observed in Mt. Gongga (mean = 3.98 ± 1.62 ng m$^{-3}$, 1sd) and Shangri-La (mean = 2.55 ± 0.73 ng


m$^{-3}$, 1sd) located on the eastern edge of the Tibetan Plateau, but much higher than that observed in
Qomolangma Natural Nature Preserve (mean = 1.42 ± 0.37 ng m$^{-3}$, 1sd) and Nam Co (mean = 1.33
± 0.24 ng m$^{-3}$, 1sd) in the inland Tibetan Plateau (Fu et al., 2012b; Fu et al., 2009; Lin et al., 2019;
Yin et al., 2018; Zhang et al., 2015a). In general, atmospheric GEM levels in remote areas are closely
related to the regional atmospheric Hg budget. The inland Tibetan Plateau is sparsely populated and
with no large-scale industrial activities. However, Some monitoring sites on the northeastern and
eastern edges of the Tibetan Plateau, such as WLG, Mt. Gongga and Shangri-La, are not too far
away from the anthropogenic Hg source regions in middle and eastern China, and thus were
impacted by anthropogenic Hg emissions through long-range transport, which explained the
relatively higher GEM concentrations at these stations than inland Tibetan Plateau stations (Fu et
al., 2008; Zhang et al., 2015a). The impact of regional and long-range transport of Hg originated
from anthropogenic emissions on the elevated GEM level at WLG was discussed in details in section
3.3 below.

Currently, there is a great debate on the measurement accuracy of GOM using KCl-coated
denuder. Therefore, GOM data in this study was only compared with previously reported data
collected using the same method. The mean GOM concentration at WLG was slightly higher than
those in rural areas of North America and China, but lower than those in urban areas in China (Fu
et al., 2012b; Zhang et al., 2016). GOM is mainly affected by local to regional emission sources and
atmospheric processes (Sheu and Mason, 2001). Since WLG is isolated from primary anthropogenic
sources, the relatively high level of GOM at WLG was probably mainly caused by atmospheric
processes. Intrusion of GOM enriched air from free troposphere could be one reason, a phenomenon
that has been reported in Qomolangma Natural Nature Preserve in the southern Tibetan Plateau (Lin
et al., 2019). Additionally, GOM at WLG generally showed relatively higher concentrations during
daytime (Fig. S1), indicating in situ photochemical production of GOM as another important
mechanism causing high GOM levels at WLG.

PBM concentrations at WLG showed large variations with the maximum hourly value reaching
655 pg m$^{-3}$. The overall mean PBM concentration at WLG (68.1 pg m$^{-3}$) was significantly higher
(by up to ~5-40 times) than those reported for remote areas in the northern Hemisphere (Kim et al.,
2012; Lan et al., 2012), but was similar to observations in the urban areas in China (Fu et al., 2012b;
Fu et al., 2015). Elevated PBM concentrations in Chinese urban areas were most likely caused by
strong local anthropogenic emissions. However, in the remote areas with few primary anthropogenic
emissions, long-range transport should be the major cause for highly elevated PBM concentrations.
PBM has an atmospheric residence time ranging from a few days to weeks and can undergo regional


transport (Seigneur et al., 2004; Zhang et al., 2019). Therefore, high PBM levels at WLG were
probably mainly caused by long-range transport of anthropogenic and natural emissions, which was
discussed in details in the following Sections.

**3.2 Seasonal and diurnal distributions of GEM, GOM and PBM**

Daily values of GEM, GOM, PBM, AT, RH, WS and RF were aggregated into monthly

average values to reveal seasonal variations during the study period (Table S1 and Fig.3). In the
discussion below, warm (May to October) and cold (November to April) seasons were compared.
Mean GEM level in the cold season (1.84 ng m$^{-3}$) was relatively lower than that in the warm season
(1.95 ng m$^{-3}$) (Table S1), which was likely due to the strengthening westerlies originated from or
passing over regions with low anthropogenic emissions during the cold season (Zhang et al., 2015b).
However, higher than seasonal-average GEM levels were observed during February to April (Table
S1), likely due to the long-range transport from northern South Asia where has been experiencing
industrialization and urbanization and thus strong anthropogenic Hg emissions (AMAP/UNEP,
2018; Chakraborty et al., 2013). Airflows originated from these areas had high GEM concentrations
and could be transported to WLG in the cold season (Lin et al., 2019; Yin et al., 2018). In addition,
higher than seasonal-average GEM concentrations were also observed during July to September
(Table S1, Fig.3), which could be attributed to the strengthening East Asia Summer Monsoon during
the warm season. The prevailing wind from the east direction (Fig.4, Table S1) could transport GEM
from eastern Qinghai and southern Gansu province of China to WLG during the East Asian Summer
Monsoon season.

Unlike GEM, mean GOM and PBM concentrations were higher in the cold than warm season

(Table S1, Fig.3), which should be mainly caused by the efficient precipitation scavenging of GOM
and PBM due to high RF in the warm season. In addition, the low RH in the cold season was
conducive to the formation of GOM and PBM through atmospheric chemical and physical
transformations (Fain et al., 2009; Lin et al., 2019). Higher PBM concentrations at WLG were
frequently detected with westerly and northerly winds (Fig. 4), which were mainly from the desert
and Gobi areas of western Tibetan Plateau, Xinjiang, southern Gansu and southwestern Inner
magnolia, suggesting that desert dust related sources in these regions could be potential sources of
PBM at WLG in the cold season.

No significant differences in GEM concentration were observed between daytime (7:00-19:00,

1.86 ng m$^{-3}$) and nighttime (20:00-06:00, 1.83 ng m$^{-3}$) at WLG (Fig.S1). This was also the case for
PBM (72.2 pg m$^{-3}$ versus 69.8 pg m$^{-3}$). The diurnal pattern of PBM at WLG was different from those


observed in Nam Co, Qomolangma Natural Nature Preserve and Mt. Gongga in the Tibetan Plateau
where PBM generally peaked during daytime under valley breeze condition (Fu et al., 2009; Lin et
al., 2019; Yin et al., 2018). The above findings at WLG suggested that local sources and in situ
atmospheric transformations may only have minor impacts on PBM concentration. Instead,
atmospheric circulation over the Tibetan plateau and long range transport from the other source
regions should be the main factors controlling the diurnal and seasonal variations of GEM and PBM
concentrations at WLG. In contrast, mean concentration of GOM during daytime (15.8 pg m$^{-3}$) was
27% higher than that during night (12.4 pg m$^{-3}$) at WLG, which was likely due to the oxidation of
GEM during the daytime (Ariya et al., 2015; Fain et al., 2009). Therefore, local meteorology and
photochemical production could be important controlling factors for the observed diurnal patterns
of GOM at WLG.

**3.3 Source identification of GEM and PBM**

During the whole study period, the prevailing winds at WLG were from southwestern quadrant

(46.5%) mainly originated from and passing over Tibetan Plateau and southern Xinjiang under the
control of the westerlies. Average GEM concentrations (1.58 to 1.91 ng m$^{-3}$) associated with this
wind sector were overall lower than those associated with other wind sectors (Fig. 4), suggesting
the areas southwest of WLG were not important source regions of GEM at WLG. In contrast, GEM
concentrations (means: 2.42 to 2.87 ng m$^{-3}$) associated with northeast wind sector were highly
elevated. The northeast wind mainly came from the low-altitude regions in northwestern China with
many anthropogenic Hg sources, which could have contributed to GEM at WLG. In contrast to
GEM, maximum PBM concentrations (means: 68.6 to 97.8 pg m$^{-3}$) were associated with wind
sectors of southwestern and northwestern quadrants and lowest PBM concentrations (means: 49.6
to 63.3 pg m$^{-3}$) were associated with wind sector of the eastern quadrants (Fig. 4). The southwest
and northwest winds were mainly originated from and passed over deserts and Gobi regions,
including the largest Taklimakan Desert in Asia. These areas are the main dust source regions in
middle and eastern China (Che et al., 2011; Chen et al., 2017), and therefore would be an important
source of PBM at WLG.

To better understand the sources and long-range transport of atmospheric Hg at WLG, CWT

values for GEM and PBM were calculated and are shown in Fig. 5. Higher GEM CWT values were
mainly located in eastern Qinghai, southern Gansu, western Shanxi, and southwestern Inner
Mongolia of China and northern South Asia, whereas lower values were mainly located western
Qinghai, Xinjiang and Xizang provinces (Fig. 5a). By matching the gridded GEM CWT values with
the gridded anthropogenic GEM emissions in the study domain, we found GEM CWT values were





significantly positively correlated with anthropogenic GEM emissions ($R^2 = 0.55$, $p < 0.01$, Fig. 6a).
This indicates GEM at WLG was mainly caused by long-range transport of anthropogenic GEM
emissions from industrial areas in western China, and this is overall consistent with the findings
discussed above that were based on wind dependence of GEM at WLG.

Differing from the case of GEM, higher PBM CWT values were mainly located in southern

Xinjiang, western Qinghai and south-central Xizang provinces, whereas the regions to the east of
WLG, where many industrial sources were located, showed relatively lower PBM CWT values (Fig.
5b). In addition, gridded PBM CWT values showed a negative correlation with gridded
anthropogenic PBM emissions (Fig. 6b). These findings indicate that long-range transport of
anthropogenic PBM emission was unlikely the major sources of PBM at WLG. Instead, long-range
transport of dust particles originated from deserts and Gobi regions in western China, such as
Taklimakan desert, Qaidam desert and Badain Jaran desert (Fig. S2), is responsible for PBM at
WLG. These regions contain the major deserts and Gobi areas in East Asia and can release up to 25
million tons dust particles annually. Dusts from these regions could be transported to the
northwestern, middle and even eastern China through the westerlies over the Tibetan plateau (Che
et al., 2011; Chen et al., 2017; Xuan et al., 2000). Previous studies showed that atmospheric PBM
concentrations (86.1-517 pg m$^{-3}$) over the Taklimakan Desert are remarkably higher than those
observed from background sites in China and even comparable to those measured in most of the
Chinese metropolitan cities (Huang et al., 2020). We thus concluded that the dry airflows transported
the PBM-enriched dust aerosols from the desert and Gobi regions to WLG, and contributed
significantly to the elevated PBM concentrations at WLG.

**3.4 Impact of desert dust related sources on PBM**

To evaluate the impact of dust related sources on the temporal variations of PBM concentration,

daily cumulative AAI (∑AAI) encountered by air masses transported during the preceding two days
were calculated, as shown in Fig. 7 together with daily PBM concentrations. PBM concentrations
maintained at relatively low levels in warm months (May to September), increased since October,
and reached the highest levels in winter and early spring (December to March) (Table S1). Daily
∑AAI showed negative values from June to October, but large positive values in winter and early
spring (Table S1 and Fig. 7). A significant positive correlation ($r^2 = 0.31$, $p < 0.01$) was observed
between daily ∑AAI and daily PBM concentration (Fig. 7), indicating that the long-range transport
of dust and/or biomass burning related sources played an important role in the temporal variations
of PBM concentration at WLG. Biomass burning related sources were not likely the major causes
because the air masses ended at WLG were mainly originated from and passed over regions with





low biomass burning area (Fig. S2). Hence, we conclude that dust related sources were the dominant
source of PBM at WLG. Previous studies analyzing spatiotemporal patterns of atmospheric dust
based on satellite remote sensing generated dust aerosol index have shown the Taklimakan area as
the dominant source of dust episodes in Asia, especially in every spring season. Desert dust is a
significant carrier of atmospheric aerosol and PBM to the cryosphere of Western China and can also
have global impact through long-range transport (Huang et al., 2020; Zhang et al., 2008).

Desert and Gobi areas are important sources of atmospheric particles. Global dust particle
emissions were estimated to range from 500 to 5000 Tg yr$^{-1}$ with an average value of 1836±903 Tg
yr$^{-1}$. In China, the desert and Gobi dust particle emissions were estimated to range from 100 to 459
Tg yr$^{-1}$ with an average value of 242±120 Tg yr$^{-1}$ (Table S2). Hg content in suspended particles from
desert dust was averaged at 0.33 μg g$^{-1}$ from existing studies (Table S2). Based on the above
numbers, PBM emissions from desert dust were roughly estimated to be $606 \pm 298$ Mg yr$^{-1}$ globally
and $80 \pm 40$ Mg yr$^{-1}$ in China. These values exceed the anthropogenic PBM emissions in the world
(75 Mg yr$^{-1}$) and China (16 Mg yr$^{-1}$), suggesting desert and Gobi areas as important sources of
atmospheric PBM emissions on regional to global scales.

Besides emissions from anthropogenic and dust related sources, gas-particle partitioning
between GOM and PBM also affect PBM level in the atmosphere. Thus, the intrusion of GOM-rich
air from free troposphere would also have an impact on PBM (Ariya et al., 2015; Lin et al., 2019;
Tsamalis et al., 2014). The PBM/GEM ratios at WLG were similar to those observed at
Qomolangma Natural Nature Preserve and Nam Co. in the Tibetan Plateau, but much higher than
those in Chinese urban and remote areas (Lin et al., 2019; Yin et al., 2018). On the other hand, the
PBM/GOM ratios at WLG were relatively lower than the values observed from the other two
Tibetan sites (Fig.8). Generally, gas-particle partitioning of GOM and PBM is mainly controlled by
air temperature (Amos et al., 2012), however, no clear dependence of monthly PBM/GOM ratio on
monthly mean air temperature was observed, e.g., similar PBM/GOM ratios were observed between
the coldest months (December to February) and other seasons (Fig. 8). Besides, air masses travelling
heights at WLG did not show clear seasonal variations throughout the study period, indicating
elevated PBM concentrations at WLG in winter and the early spring were unlikely associated with
intrusions of free troposphere air masses. We thus conclude that gas-particle partitioning of GOM
was not likely the major cause of the elevated PBM at WLG.

**4 Conclusions**
This study presented the first full-year continuous speciated Hg data set and identified potential



350 sources causing high GEM and PBM at WLG in the Tibetan Plateau. Mean GEM level at WLG was

351 slightly higher than the background level of GEM in the Northern Hemisphere. Mean PBM level at

352 WLG was much higher compared with the reported values in remote areas in the Northern

353 Hemisphere. Seasonal variations in GEM concentration indicated that Hg emissions from

354 anthropogenic source regions and long-rang transport played important roles on the high GEM

355 levels at WLG. High PBM concentrations at WLG were observed in cold season, which were mainly

356 caused by dust aerosol sources from the desert and Gobi areas. Analysis from CWT and ∑AAI

357 northern Xinjiang, eastern Qinghai, southern Gansu, southwestern Shaanxi, western Inner Mongolia

358 of China and northern South Asia could be the main source areas of GEM, while southern Xinjiang,

359 southwestern Inner Mongolia, northern Gansu, western Qinghai and Tibet of China were likely the

360 source regions of PBM at WLG. Long-range transport of dust particles from desert and Gobi areas

361 contribute to the elevated PBM at WLG. The estimated PBM emissions from dust particles

362 suggested that dust from desert and Gobi areas are critical sources of PBM on regional to global

363 scales, which should be paid more attention in future studies.

364

**Acknowledgments**

366 This work is supported by the Strategic Priority Research Program (XDB40000000), Key

367 Research Program of Frontier Sciences (ZDBS-LY-DQC029) of CAS, the National Science

368 Foundation of China (41703134 and 41622305), and the K.C. Wong Education Foundation of CAS.

369 We also thank staffs in Waliguan GAW Baseline Observatory for field sampling assistance.

370

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



**Figure Captions**

**Fig. 1:** The map showing the location of WLG, and distributions of the deserts and cities around WLG.

**Fig. 2:** Time series of hourly GEM, GOM and PBM concentrations in ambient air at WLG.

**Fig. 3:** Monthly means of GEM, GOM and PBM at WLG during a full-year sampling period.

**Fig. 4:** Frequency distribution of wind direction and wind-sector based mean GEM and PBM concentrations during the study period.

**Fig. 5:** Identified source regions of atmospheric GEM and PBM at WLG during the study period. (A) gridded (0.5°×0.5°) values for GEM, and (B) gridded (0.5°×0.5°) values for PBM. Gray line enclosed regions indicate desert locations in China.

**Fig. 6:** Correlation between the simulated GEM or PBM CWT value and their respective anthropogenic emissions. The location of each gridded CWT value is matched with that of anthropogenic emission. Gridded anthropogenic emissions are divided into 20 groups with equal number of grids, starting from the lowest to highest emission values.

**Fig. 7:** Variations in daily mean PBM concentration and daily cumulative Absorbing Aerosol Index (AAI) during the proceeding two days at WLG.

**Fig. 8:** The monthly PBM/GEM and PBM/GOM ratio with air temperature and air mass traveling height during a full-year sampling period. Columns (a), (b) and (c) are the mean PBM/GEM and PBM/GOM ratios at Qomolangma Natural Nature Preserve and Nam Co in the inland Tibetan Plateau, Chinese cities and Chinese remote areas, respectively.




**Fig. 1:** The map showing the location of WLG, and distributions of the deserts and cities around
WLG.

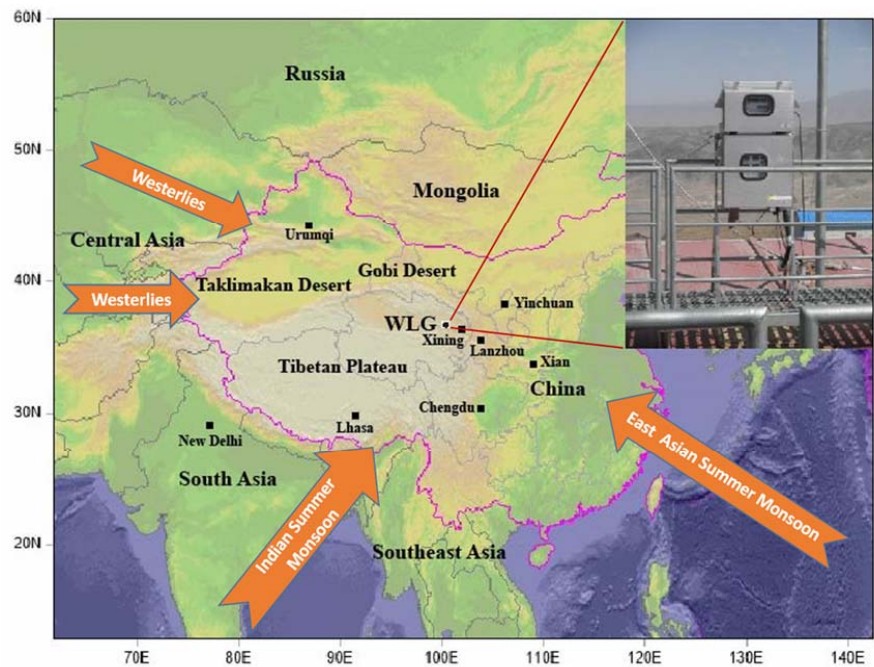





594 **Fig. 2:** Time series of hourly GEM, GOM and PBM concentrations in ambient air at WLG.

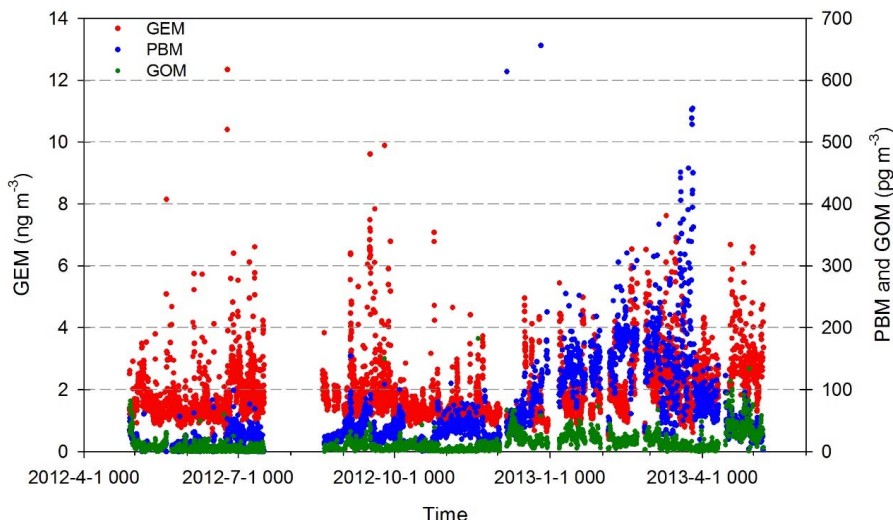







**Fig. 3:** Monthly means of GEM, GOM and PBM at WLG during a full-year sampling period.

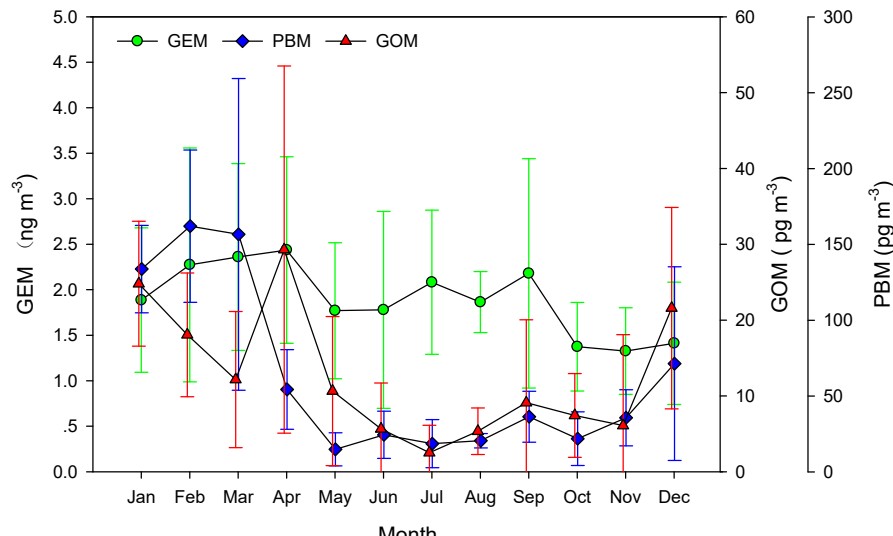






**Fig. 4:** Frequency distribution of wind direction and wind-sector based mean GEM and PBM concentrations during the study period.

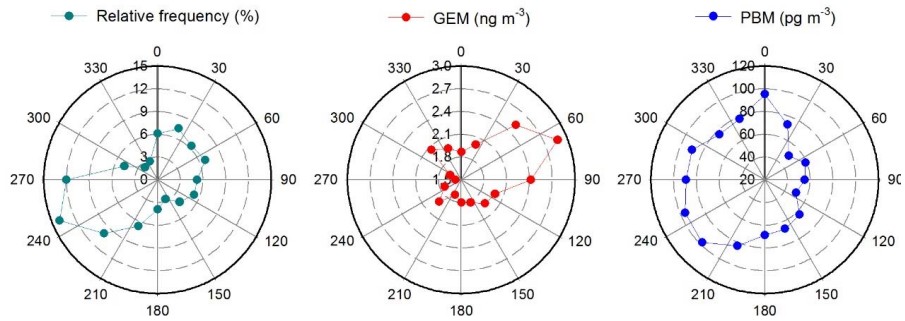





**Fig. 5**: Identified source regions of atmospheric GEM and PBM at WLG during the study period.
(A) gridded (0.5°×0.5°) CWT values for GEM, and (B) gridded (0.5°×0.5°) CWT values for PBM.
Gray line enclosed regions indicate desert locations in China.

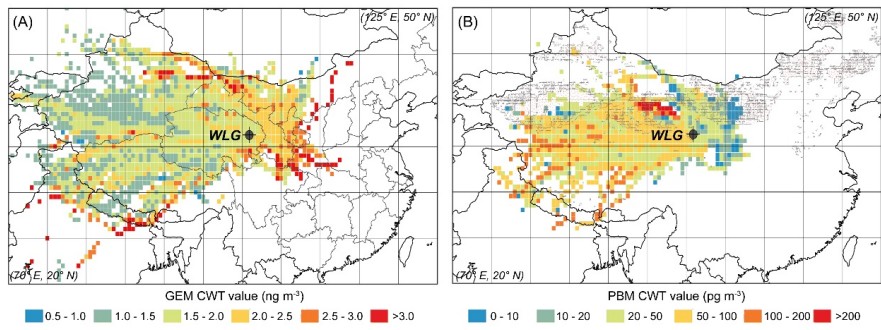






**Fig. 6:** Correlation between the simulated GEM or PBM CWT value and their respective anthropogenic emissions. The location of each gridded CWT value is matched with that of anthropogenic emission. Gridded anthropogenic emissions are divided into 20 groups with equal number of grids, starting from the lowest to highest emission values.

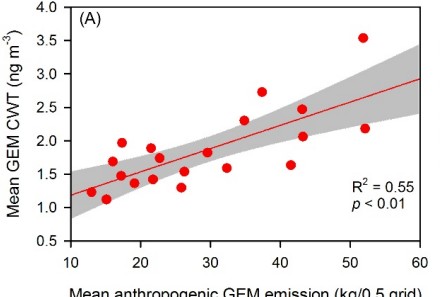

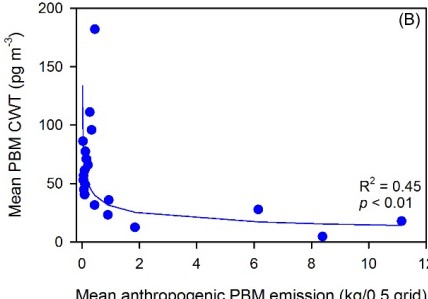





**Fig. 7:** Variations in daily mean PBM concentration and daily cumulative Absorbing Aerosol Index
(AAI) during the preceding two days at WLG.

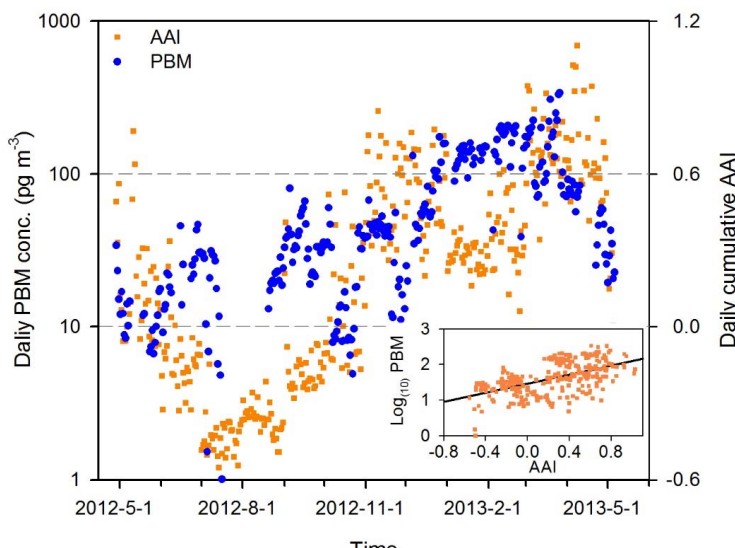





**Fig. 8:** The monthly PBM/GEM and PBM/GOM ratio with air temperature and air mass traveling
height during a full-year sampling period. Columns (a), (b) and (c) are the mean PBM/GEM and
PBM/GOM ratios at Qomolangma Natural Nature Preserve and Nam Co in the inland Tibetan
Plateau, Chinese cities and Chinese remote areas, respectively.

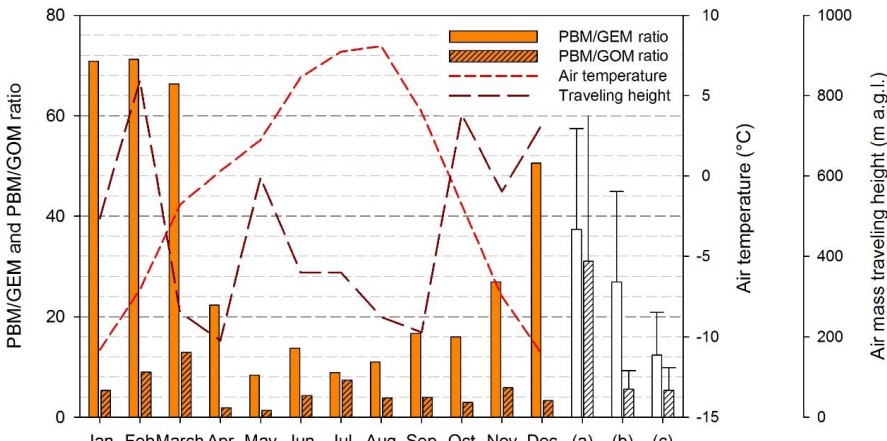


