# Peer review of "Speciated atmospheric mercury at Waliguan Global Atmospheric Watch station in the northeastern Tibetan Plateau: implication of dust related sources for particulate bound mercury"

_Atmospheric Chemistry and Physics, 2021_

## Author Comment (AC1)

**Response to Referee #1**

RC- Reviewer's Comments; AC – Authors' Response Comments

**RC1:** This paper reports a full-year continuous measurement of speciated atmospheric mercury at Waliguan Baseline Observatory, which is a valuable addition to our understanding of atmospheric mercury cycle especially as this site is an important background site for many contaminants. The data quality reported here is quite high and the related analysis is also appropriate and deep. The paper is generally well written with high-quality figures, and the discussions about PBM and dust levels are also interesting and novel. I suggest publishing this paper after a minor revision.

**AC1:** We appreciate Prof. Zhang for reading our manuscript and providing constructive comments. We have studied these comments carefully and made corrections accordingly, which have been marked in blue in the revised manuscript. The response to these comments are presented below.

Specific comments:

**RC2:** Line 26: Can GEM also be adsorbed by aerosols?

**AC2**: Yes, GEM can be adsorbed by atmospheric aerosols to form PBM, and the gas-partitioning equilibria is expected to be controlled by Henry's law (Ariya et al., 2015).

**RC3:** Line 280: Not clear what each point in Figure 6a represents, a group of 0.5x0.5 grid cells? How do you aggregate them? Which emission inventory do you use?

**AC3**: We have specified these information in the caption of Figure 6 in the revised manuscript.
Yes, each point in Figure 6a represents a group of CWT values (or anthropogenic emission) in 0.5×0.5 grid cells. As shown in the caption of Figure 6, gridded anthropogenic emissions in all grid cells (0.5°×0.5°) in the study domain were ranked starting from the lowest value to the highest value, and then the gridded anthropogenic emissions were divided into 20 groups with equal number of grid cells, starting from the lowest to the highest. The average anthropogenic emissions as well as the matched GEM and PBM CWT values in these 20 groups were then calculated to conduct the relationship analysis.
Anthropogenic emission data are from AMAP/UNEP, 2018.

**RC4:** Line 285-301: The observed high PBM concentrations are caused by the high dust load? I suggest reporting the dust concentration levels at Waliguan as well, and a comparison with urban sites. Also, the instrument only measures PBM minus than 2.5 micron, but the dust is probably mainly in the coarse mode. How to reconcile this discrepancy?

**AC4**: The suggestion of further analysis between PBM concentrations and dust concentrations are appreciated. Unfortunately, we are not able to collect the dust concentrations datasets at WLG. Absorbing Aerosol Index (AAI) is one of the most useful space-borne data sets, which dominantly

presents the levels of desert dust and biomass burning related aerosols (biomass burning could be neglected in the Tibetan Plateau, more detail please see Figure S5). Therefore, we deem that the use of AAI is suitable to investigate the effect of dust related sources on PBM in the study area.

For the difference regarding the aerodynamic diameter between PBM and dust aerosols, previous studies observed that the $PM_{2.5}$ concentration in the dust is significantly positively correlated with the concentrations of TSP and $PM_{10}$ concentration (Fadi A. Al-Jallad et al., 2017; Hemraj Bhattarai et al., 2021). Therefore, the total concentrations of dust aerosols, which is highly linked with the $PM_{2.5}$ concentrations within the dust aerosols well, could be useful to investigate the impact of dust sources on PBM at WLG.

**RC5:** Line 322-325: It's also important to specify the size range of these dust emission estimates.

**AC5**: good point, the ranges are added in line 345 and 347 in the revised manuscript, which reads: "*total particulate bound mercury (Hg-TSP) emissions from desert dust related sources were roughly estimated to be $606 \pm 298$ (range from 165 to 1650) Mg $yr^{-1}$ globally and $80 \pm 40$ (range from 33 to 151) Mg $yr^{-1}$ in China (Table S3).*"

**RC6:** Line 325-330: I would also refrain from suggesting an "emission flux" from the suspended dust particles, as the Hg on particles may be from adsorbing from the ambient atmosphere during transport after release.

**AC6**: we understand reviewer's concerns. Actually, our estimate represents the sum of PBM emitted from primary desert sources and PBM formation during subsequent atmospheric processes (e.g., adsorption of gaseous Hg). Currently, we are not able to calculate the exact values of these two potential sources. In the revised manuscript, referred these two sources as to "desert dust related sources".

**RC7:** Line 332-346: Both temperature and particulate matter (PM) load influence the PBM/GOM ratio. Have you checked the relationship between temperature and the PBM/PM/GOM? I guess if you normalize the PBM/GOM ratio by the particulate matter load, you would get a more consistent results with other Tibetan sites.

**AC7**: we agree with the reviewer the gas-particle portioning of gaseous Hg could be controlled by air temperature and the concentrations of particulate matter. The point that we would like to show here is that the elevated PBM concentrations in winter and early spring at WLG was not likely caused by the enhanced gas-particle portioning of GOM under their low air temperature, and we have revised this part in line 364-366 in the revised manuscript.

**Reference used in this response letter**

1. AMAP/UNEP, 2013. Geospatially Distributed Mercury Emissions Dataset 2010v1.

2. Ariya, P.A., Amyot, M., Dastoor, A., Deeds, D., Feinberg, A., Kos, G., Poulain, A., Ryjkov, A., Semeniuk, K., Subir, M., Toyota, K. (2015) Mercury Physicochemical and Biogeochemical Transformation in the Atmosphere and at Atmospheric Interfaces: A Review and Future Directions.

Chemical Reviews 115, 3760-3802.

3. Fadi A. Al-Jallad, Clarence C. Rodrigues, Hamda A. Al-Thani: Ambient Levels of TSP, PM10, PM2.5 and Particle Number Concentration in Al Samha, UAE, Journal of Environmental Protection, 8, 1002-1017, 2017.

4. Hemraj Bhattarai, Lekhendra Tripathee, Shichang Kang, Pengfei Chen, Chhatra Mani Sharma, Kirpa Ram, Junming Guo, Maheswar Rupakheti: Nitrogenous and carbonaceous aerosols in PM2.5 and TSP during pre-monsoon: Characteristics and sources in the highly polluted mountain valley, journal of environmental sciences 115, 10–24, 2021.

---

## Author Comment (AC2)

**Response to Referee #2**

**RC**- Reviewer's Comments; **AC** – Authors' Response Comments

**RC1:** The authors present one year of measurements of speciated mercury at a mountain station Walliguan (WLG). They interpret the data in terms of diurnal and seasonal variations. Using concentration weighted trajectory approach they identify source regions. Desert dust has been identified a significant source of particle bound mercury (PBM).

The paper is well ordered mostly well written. Unfortunately, the data analysis is rather superficial leading to sometimes questionable conclusions. The major problem is the frequently occurring pollution events combined with a statistical analysis based on averages and their standard deviations which will be substantially influenced by the pollution events. The pollution events are not discussed although their backward trajectories and chemical signatures could provide additional information about the sources. The data are valuable and deserve to be published, after some improvement of their analysis.

**AC1:** We appreciate the reviewer for dedicating time to review our manuscript and provide constructive comments. All the comments are appreciated, and we have revised manuscript following the comments.
Regarding the statistical analysis, we have added the median values in the revised manuscript to avoid comparison artifact between this and previous studies.
The pollution events were also investigated in the revised manuscript. Please see the authors' response below.

**General comment:**

**RC2:** To the best of my knowledge, WLG is a WMO GAW station and many other trace gases and aerosol parameters are being measured there, in addition to Hg and its speciation. I wonder, why only one or two of these in-situ measurements is used for the interpretation of the speciated Hg measurements. Seasonal variation of directly measured dust concentrations would be useful e.g. for the discussion in the section 3.3 and 3.4. The use of these measurement would substantiate the findings and the conclusions of the paper.

**AC2**: We thank the reviewer for the suggestion of using criteria pollutant parameters to interpret our dataset. We have collected CO and BC (black carbon) dataset at WLG (Table S1). These two pollutants are dominantly derived from anthropogenic activities and could be used to investigate whether our observations were mainly impacted by anthropogenic activities or not.
We conducted the relationship analysis between speciated atmospheric mercury associated and CO and BC concentrations (Table S2), and the result is used to support our hypothesis that PBM pollutions at WLG were not likely caused by anthropogenic sources, and more details please see line 331-334, which reads: "*PBM concentrations showed a negative correlation with CO and BC concentrations, which are mainly emitted from the industrial and biomass burning activities (Table*

*S2). Hence, we conclude that the dust related sources were the dominant source of PBM at WLG.*".

**RC3:** RH is inversely related to AT, and thus essentially redundant to it. Air water content, which can be easily calculated from RH and AT, would be a really independent parameter and thus a preferable one.

**AC3**: Good point. We have calculated the monthly mean air water contents and add them in Table S1.

**RC4:** Figure 2 shows numerous pollution events with seasonally varying frequency of their occurrence. Consequently, discussion in terms of averages will blur the differences because of insignificant differences due to large standard deviations. Medians or seasonal and diurnal event frequencies could provide a more transparent insight as would an analysis of event frequencys.

**AC4:** Good point. The annual median values of speciated atmospheric Hg at WLG are presented in line 178 in the revised manuscript. The monthly and daily median values of speciated atmospheric Hg at WLG are presented in Table S1 and Figure S1. We found the monthly median and mean values of GEM, PBM and GOM showed quite similar seasonal and distribution patterns (Table S1), and the high-temporal variations in GEM and PBM depicted by daily mean and median values are also consistent.

**RC5:** Section 2.2 has a subsection 2.2.1 but no subsection 2.2.2?
**AC5**: Revised.

**RC6:** Section 2.2.1: GEM detection limit of 0.1 ng m$^{-3}$ is given, but what are the GOM and PBM detection limits? Please provide sampling flow rates and sampling durations for GEM, GOM and PBM. The problem is that with the usual 5 min and 1 l/min for GEM and 2 h with 10 l/min for GOM and PBM not enough mercury is collected for unbiased and precise analysis by Tekran (Ambrose, Atmos. Meas. Tech., 10, 5063-5073, 2017; Slemr et al., Atmos. Meas. Tech., 9, 2291-2302, 2016). The information about sampling intervals and flow rates is thus necessary to assess the accuracy and precision of the presented measurements. Because of the high altitude of WLG it should be also stated whether the concentrations are related to m$^3$ at standard pressure and temperature.

**AC6**: The sampling flow rates and intervals are presented in line 113-118 in the revised manuscript, which reads: "*Due to the low air pressure at WLG, the total sampling flow rate of the GOM and PBM was programed to be 6.6 lpm (referenced to standard temperature and pressure conditions). The Tekran 2537 was sampling GEM at a flow rate of 0.6 lpm, while the Tekran 1130 pump module pulled additional air at 6 lpm. A 2-hour duration was selected for GOM and PBM sampling, during which GEM is continuously measured at a 5-minute interval.*"

The detection limits of GOM and PBM are presented in line 109-110, which reads:" *The typical detection limits for the GOM and PBM measurements during a 2-hour sampling duration are 2 pg m$^{-3}$, respectively*".

The analytical uncertainty of GOM caused by small Hg load during instrumental integration is shown in line 198-199, which reads:" *and a small load of Hg could also cause analytical uncertainties in Tekran-based GOM and PBM measurements (Ambrose, 2017).*"

**RC7:** Section 2.2: Backward trajectories were calculated every 4 h. Presumably GEM, GOM, and PBM were averaged over the same time stamp but this is not mentioned in the text.

**AC7**: Yes, GEM and PBM concentrations were averaged to be the 4-h means to match the calculated trajectories, which is specified in line 142-143.

**RC8:** Section 3.1: Because of the GEM temporal trends, GEM measured at WLG in 2012 and 2013 should be preferably compared with measurements at other sites made in the same years. Figure 2 shows frequent pollution events which are not mentioned in the GEM discussion. They will drive the averages and standard deviations up, medians would provide a more representative information.

**AC8**: In the present study, we mainly compare our observations with previous studies conducted during 2011-2015. Therefore, the comparation between WLG and other sites in the Northern Hemisphere should be relevant.
The pollution events of GEM, PBM and GOM were introduced in line 231-233, which reads:" *However, elevated monthly mean GEM levels were observed from February to April (Table S1, Fig. S1, Fig. S2), and many high GEM events were frequently observed in the cold season (Fig. S3)",* and in line 244-245, which reads:" *Also, the high GOM and PBM events occurred mainly in the cold months (Fig. S3)*".

**RC9:** Line 186: ..will be discussed in detail…

**AC9**: Revised.

**RC10:** Paragraph starting at line 189: The problem with the internal Tekran signal integration mentioned above is another reason for low bias of GOM measured by the KCl denuder. As such it should be mentioned here too.

**AC10**: Yes, this potential analytical artifact is mentioned in line 198-199 in the revised manuscript.

**RC11:** Section 3.2: Because of the frequent pollution event the discussion here in terms of averages is obscure. A discussion of monthly event frequencies would provide a more transparent insight. E.g. pollution events are much more frequent in the cold season when compared with the warm one.

**AC11**: Good point. A plot regarding the monthly pollution events of GEM. GOM, and PBM are shown in Fig.S3. Discussions regarding these speciated atmospheric Hg pollution events are also added in the revised manuscript. More details please see the response to the comments RC8.

**RC12:** Paragraph starting at line 232: "… low RH in the cold season was conducive to the formation of GOM and PBM..". Cold season (November – April) is essentially winter, i.e. GOM and PBM according to this finding are more efficiently produced in winter. This is at odds with observations of wet Hg deposition peaking in summer almost everywhere (e.g. Cole et al., Atmosphere, 5, 635-668, 2014).

**AC12**: No, our interpretation agree with previous observations of wet Hg depositions. We have clarified this statement in line 245-250 in the revised manuscript, which read:" *Lower GOM and PBM concentrations in the warm season were probably attributed to the increasing removal processes of these water soluble Hg species, and this is consistent with previous observations with wet Hg deposition fluxes peaked in the warm rainy season (Cole et al., 2014). In addition, low RH in the cold season would be conducive to the formation of GOM and PBM through atmospheric chemical and physical transformations (Fain et al., 2009; Lin et al., 2019).*"

**RC13:** Paragraph starting at line 242: The given numbers without the standard deviations and the number of measurements do not allow to judge whether there is a difference between day and night. In addition, because of seasonal GOM and PBM variations the diurnal variations should be investigated separately for different seasons.

**AC13**: Standard deviations of concentrations measured during daytime and nighttime are shown in line 257-258 in the revised manuscript.
We also analyzed the diurnal variations of GOM and PBM in warm and cold season respectively, which are similar to that of the whole sampling period. For example, there was no significant differences in PBM concentration between daytime and nighttime (warm season: $63.1\pm74.9$ ng m$^{-3}$ versus $52.6\pm65.7$ ng m$^{-3}$; cold season: $93.4\pm82.4$ ng m$^{-3}$ versus $93.8\pm71.7$ ng m$^{-3}$). Mean concentration of GOM during daytime was slightly higher than that during night (warm season: $13.0\pm12.2$ ng m$^{-3}$ versus $9.7\pm8.2$ ng m$^{-3}$; cold season: $20.8\pm18.4$ ng m$^{-3}$ versus $16.1\pm11.0$ ng m$^{-3}$). Since the diurnal variations of all Hg species were similar during cold and warm seasons, we did not conduct further analysis in the diurnal variations in different seasons.

**RC14:** Lines 251-255: It is generally very difficult to separate chemistry from transport in diurnal variations without specific tracers because of diurnal PBL dynamics. It is even more complicated at mountain stations with additional upslope and downslope winds, see e.g. Weiss-Penzias et al. (J. Geophys. Res., 111, D24301, doi:10.1029/2006JD007415, 2006). The attribution of diurnal variation to chemistry here is also highly questionable for another reason: With mercury lifetime of $0.5 – 2$ yr, mentioned in the introduction, the day/time difference should be nondetectable considering the GOM standard deviations reported here.

**AC14**: We agree with the reviewer that the alternations of upslope and downslope winds wound have a potential impact on the diurnal variations in atmospheric Hg. Generally, upslope which carry boundary polluted air would drive an increase of atmospheric Hg at many mountainous sites. This, however, is quite different at WLG because there is no significant point sources in the surrounding low-altitude areas. We have revised this paragraph to strength our hypothesis in line 267-269, which reads:" *In contrast, mean concentration of GOM during daytime (17.2 ± 16.5 pg m$^{-3}$) was 31.3%*

*higher than that during night (13.1 ± 10.3 pg m⁻³) at WLG. Given that there was a lack of strong anthropogenic emissions around the station or in the surrounding areas, such a daytime elevated GOM phenomenon should be likely attributed to the in situ production of GOM via GEM oxidation during the daytime*".

**RC15:** Section 3.3: Why is GOM omitted from the discussion?

**AC15**: The reasons are shown in line 197-200, which reads:" Currently, there is a great debate on the measurement accuracy of GOM using KCl-coated denuder, and a small load of Hg could also cause analytical uncertainties in Tekran-based GOM and PBM measurements (Ambrose, 2017). Therefore, GOM data in this study was only compared with previously reported data collected using the same method.".

**RC16:** Paragraph starting at line 332: It is true that gas-particle partitioning is mainly controlled by temperature. At WLG, however, it will be to a large degree controlled also by the available aerosol surface area which is probably orders of magnitude larger in air masses transported from the desert when compared with other air masses. Measured dust concentrations from the GAW monitoring at WLG could provide a better insight in the seasonal variation of PBM/GOM ratio.

**AC16**: We agree that the gas-particle partitioning could be controlled by air temperature and aerosol concentrations. The point we would like to convey is that elevated PBM concentrations in winter at WLG is not due to the low air temperature, which may enhance gas-particle portioning of gaseous Hg. We have rephrase this statement in line 364-366, which reads:" *no clear dependence of monthly PBM/GOM ratio on monthly mean air temperature was observed, e.g., similar PBM/GOM ratios were observed between the coldest months (December to February) and other seasons (Fig. 8). This indicates the elevated PBM in winter and early spring at WLG were not likely caused by the enhanced gas-particle partitioning of GOM under low air temperature.*".
We are not able to collect aerosol concentrations at WLG. The reviewer point is consistent with our major explanation throughout the manuscript, that is primary desert aerosol release as well as subsequent transformation between gaseous Hg and durst aerosols (the sum could be referred as to dust related sources) contributed significantly to the PBM pollutions at WLG.

**RC17:** Figure 3: What is the meaning of the bars: standard deviations? Monthly medians would provide a more representative seasonal variation, at least for GEM. Alternatively, seasonal variation of pollution event frequencies should be discussed because it determines the monthly averages and their standard deviation.

**AC17:** Yes, bars indicate the 1sd. Monthly median values are provided in Table S1. Pollution events as well as the related interpretation are also presented in the revised manuscript. Please see more details in AC8.

**RC18:** Figure 8: The caption is confusing: with the ratios at Qomolangma Nam Co, Chinese cities and Chinese remote areas one would expect an additional column d because urban and remote areas are probably different in PBM/GEM and PBM/GOM ratios?

**AC18**: Thanks for pointing out this. We have revised the caption of Fig 8 to avoid confusions.

**RC19:** Figure S1: The RH curve without advection should essentially mirror the AT curve, i.e. it should peak at AT minimum and vice versa. The deviation from this idealised relation shows the diurnal change of local transports. Such transports of different air masses prevent the attribution of diurnal variations solely to chemistry.

**AC19**: Thanks for this important knowledge. The comments here is similar to RC14, and we have made response (AC14) to RC14.

---

## Author Response (AR2)

**Response to Referee #2**

**RC**- Reviewer's Comments; **AC** – Authors' Response Comments

**RC**:Line 266: Is the difference statistically significant?

**AC:** We appreciate the reviewer for reviewing our manuscript again and providing the comment. The difference in GOM concentrations between daytime and night at WLG is statistically significant, and the statistical analysis result is shown in line 265 in the revised manuscript, which reads: "*($p < 0.01$, Two-independent sample t-test)*".